# A Numerical Model for Enzymatically Induced Calcium Carbonate Precipitation

**Johannes Hommel** [1,*], **Arda Akyel** [2], **Zachary Frieling** [2], **Adrienne J. Phillips** [2],
**Robin Gerlach** [2], **Alfred B. Cunningham** [2] **and Holger Class** [1]

1   Department of Hydromechanics and Modelling of Hydrosystems, University of Stuttgart, Pfaffenwaldring 61, 70569 Stuttgart, Germany; Holger.Class@iws.uni-stuttgart.de
2   Center for Biofilm Engineering, Montana State University, 366 Barnhard Hall, Bozeman, MT 59717, USA; arda.akyel@montana.edu (A.A.); zachary.frieling@gmail.com (Z.F.); adrienne.phillips@montana.edu (A.J.P.); robin_g@montana.edu (R.G.); al_c@montana.edu (A.B.C.)
*   Correspondence: johannes.hommel@iws.uni-stuttgart.de; Tel.: +49-711-685-64600

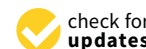

**Featured Application:** **Enzymatically induced calcium carbonate precipitation can be used for all types of subsurface engineered permeability modification such as wellbore leakage mitigation or enhanced oil or gas recovery. The model developed in this study can be used to design and assess such applications.**

**Abstract:** Enzymatically induced calcium carbonate precipitation (EICP) is an emerging engineered mineralization method similar to others such as microbially induced calcium carbonate precipitation (MICP). EICP is advantageous compared to MICP as the enzyme is still active at conditions where microbes, e.g., *Sporosarcina pasteurii*, commonly used for MICP, cannot grow. Especially, EICP expands the applicability of ureolysis-induced calcium carbonate mineral precipitation to higher temperatures, enabling its use in leakage mitigation deeper in the subsurface than previously thought to be possible with MICP. A new conceptual and numerical model for EICP is presented. The model was calibrated and validated using quasi-1D column experiments designed to provide the necessary data for model calibration and can now be used to assess the potential of EICP applications for leakage mitigation and other subsurface modifications.

**Keywords:** reactive transport; induced mineral precipitation; biomineralization; porosity and permeability reduction; leakage mitigation

## 1. Introduction

Enzymatically induced calcium carbonate precipitation (EICP) occurs when the activity of an enzyme alters the surrounding aqueous phase leading to precipitation of calcium carbonate. In this study, we focus on EICP via ureolysis by the enzyme urease of the bacterium *Sporosarcina pasteurii*, which is known for producing high amounts of urease. The enzyme urease catalyzes the hydrolysis reaction of urea ($(NH_2)_2CO$) into ammonia ($NH_3$) and carbon dioxide ($CO_2$). The ureolysis reaction leads to an increase in pH as aqueous solutions of ammonia become alkaline. This results in higher concentrations of dissolved carbonate ($CO_3^{2-}$) as the dominant inorganic carbon species at high pH conditions. In the presence of calcium ($Ca^{2+}$), this results in the precipitation of calcium carbonate ($CaCO_3$). The overall EICP reaction is as follows:

$$(NH_2)_2CO + 2H_2O + Ca^{2+} \longrightarrow 2NH_4^+ + CaCO_3 \downarrow \tag{1}$$

Similar to microbially induced calcium carbonate precipitation (MICP), EICP offers an engineering strategy to precipitate calcium carbonate in situ to change soil parameters such as mechanical strength, porosity, and permeability. As a technology, it can be used similarly to MICP to alter hydraulic flow conditions or for soil stabilization [1–6]. EICP has already been applied for soil stabilization [7]. It could potentially also be used for building or monument restoration [8], and heavy metal coprecipitation [9,10]. The capability of induced precipitation to seal highly permeable leakage pathways has been demonstrated for MICP in various studies [11–17].

Successful application of EICP however depends on the interplay between the transport of urease as well as urea and calcium, determined by fluid dynamics, sorption of the urease to solid surfaces, and the reaction rates. Due to this interplay, the predictive planning of ureolysis-induced calcite precipitation and its impact is a major difficulty for practical engineering applications. One step towards overcoming this difficulty is the numerical model that we provide for EICP. For MICP, numerical models have been shown to be capable of capturing the complex interplay of hydraulics, precipitation reactions, and the change in hydraulic properties [12,18–20].

Numerical modeling is also an appropriate tool to assess the various emerging induced precipitation methods, providing the possibility to investigate each method's strong or weak points for a given setup to assist in the choice of the most suitable method for a given goal. After the choice of the specific method, models can further assist the application of induced precipitation methods, evaluating potential injection strategies or supporting the monitoring of an application by complementing sparse experimental measurements or by providing estimates for parameters and processes which are difficult to measure. This is of increasing importance as more and more induced precipitation methods are developed and applied.

## 2. Relevant Processes and Experimental Data

Column experiment data are used for model calibration and validation using inverse modeling. The column studies were performed similar to those described for MICP in [21]. All columns were constructed from a polyvinyl chloride (PVC) pipe of 2.43 cm inner diameter and 61 cm length filled with 40 mesh quartz sand (0.5 mm effective filtration size, Unimin Corp., Emmet, ID, USA), packed under water, and vertically positioned. For the first column (#1), the mineralization medium included equimolar concentrations of urea and $Ca^{2+}$ at 0.33 M, which was doubled for the second column (#2, 0.66 M). The columns had initial pore volumes of 97.7 mL and 97.3 mL, respectively. Both columns were preheated to 60 °C by placing them in a temperature-controlled oven and kept in the oven for the course of the experiments. Sixty degrees Celsius was determined to be the temperature at which the optimal rate of ureolysis was achieved for the bacterial and plant-based sources of urease under conditions similar to those in the column experiment [22,23]. This temperature is a compromise between a fast ureolysis rate and a relatively low urease inactivation rate, both rates increasing exponentially with temperature according to an Arrhenius-type relationship. A temperature of 60 °C thus maximizes the ureolytic activity in the experimental column. *S. pasteurii* cells were used as the source of urease; the bacterial cells became thermally inactivated quickly at 60 °C and were not culturable on agar plates. This is in agreement with other reports of *S. pasteurii* not being able to grow above 40 °C [24]. However, while plant-based and *S. pasteurii* ureases are inactivated at 60 °C, the inactivation is slow enough to allow for significant ureolysis to occur [22,23]. Thus, ureolysis in this experiment was catalyzed predominantly by the residual urease of the inactivated cells, while initially, to some extent, living-cell urea hydrolysis might have occurred although that was not measured.

The injection strategy for column #1 consisted of a two-pore-volume injection of *S. pasteurii* cell suspension with an optical density (OD) of $0.85 \pm 0.01$ at 600 nm wavelength (OD 600 nm), measured in

96-well flat-bottom plates using 200 μL of culture volume (the background blank was OD 600 nm = 0.04). An OD 600 nm of $0.85 \pm 0.01$ was found to be equivalent to biomass concentrations of $0.93 \pm 0.02$ g/L cell dry weight, determined through filtration of a cell suspension in triplicate, rinsing with sterile water and drying to constant weight at 45 °C. The cell suspension of OD 600 nm = 0.85 hydrolyzed 90% of urea from the mineralization solution within 45 min at 60 °C for the urea concentrations used in the experiments, while at 30 °C, 240 min was required. Cell-suspension injection was followed by 10 mL of a spacer solution of 10 g/L $NH_4Cl$ and a two-pore-volume injection of mineralization medium with a 0.33 molar concentration of urea and calcium. For column #2, the injection strategy was similar to that of column #1 but *S. pasteurii* cell suspension and spacer were injected only before every odd-numbered mineralization-medium injection for which the concentration was doubled to 0.66 M. The flow rate was for both columns 54.4 mL/min $= 9.06667 \times 10^{-7}$ m³/s from bottom to top. Each sequence of media injection was followed by a two-hour batch period without injection, after which the injection sequence was repeated for a total of 13 times for column #1 and 12 times for column #2. For column #2, the full injection sequence of cell suspension, spacer, and mineralization medium was injected only during the odd-numbered injections. During the even-numbered injections, only the mineralization medium was injected into column #2. Thus, column #1 received 13 cell-suspension and mineralization-medium injections, while column #2 received 6 cell-suspension and 12 mineralization-medium injections. There were additional 15.52 h overnight resting periods after injections 4 and 8.

The mineralization medium, together with the spacer solution, was stored inside the oven at 60 °C to minimize non-isothermal effects. The *S. pasteurii* cell suspension was stored at room temperature, to minimize inactivation of cells and urease through prolonged exposure to high temperatures. Cell suspensions were injected without preheating into the tubing leading to the column. The tubing volume of the cell-suspension injection line was 54 mL, half of which was inside the oven, allowing some heating before the cell suspension entered the column. The minimum temperature measured with a thermocouple (OM-EL-USB-TC-LCD, OMEGA, Norwalk, CT, USA, with the probe THS-113-373-T-L, ThermoWorks, American Fork, UT, USA) at the column was 58 °C during the cell-suspension injection.

The columns were constructed with two sampling ports at 10.16 cm and at 40.64 cm. Samples were taken during each batch period at 1, 15, 30, 60, and 120 min after each previous mineralization medium injection for column #1 (0.33 M) and at 1, 15, 30, 45, 60, 90, and 120 min for column #2 (0.66 M). Approximately 1 mL of sample was extracted using a syringe and needle and filtered through a 0.2 μm filter. Sixty microliters (60 μL) of each sample were transferred immediately into a centrifuge tube containing 540 μL of 0.56 M $H_2SO_4$ and stored at 4 °C to stop any ureolytic activity until analysis. All samples were analyzed for their urea and $Ca^{2+}$ concentrations using the colorimetric assays by [25,26], for urea and calcium, respectively, modified as described in [27,28]. At the end of the experiment, the column was destructively sampled by cutting it into twelve 2-inch sections. Triplicate samples of approximately 1 g of each section were digested with 10% trace-metal-grade nitric acid (Fisher, Fair Lawn, NJ, USA). The resulting $Ca^{2+}$ concentration of the solution was measured using the colorimetric assay [26], modified as described by [28]. The remaining solids after acid digestion were rinsed and dried to determine the mass of sand from which $Ca^{2+}$ had been dissolved. The experimental results are shown in Tables A1 and A2 and are available at https://darus.uni-stuttgart.de/dataverse/sfb1313_eicp_model_calibration.

Enzymatic ureolysis is the driving force of EICP hydrolyzing urea. At typical environmental pH conditions, ureolysis produces bicarbonate ($HCO_3^-$) and ammonium ($NH_4^+$):

$$(NH_2)_2CO + 3\,H_2O \xrightarrow{\text{urease}} NH_4^+ + H_2NCOO^- + 2\,H_2O \longrightarrow 2\,NH_4^+ + HCO_3^- + OH^- \qquad (2)$$

Due to the production of hydroxide (OH⁻), ureolysis leads to an alkalinization of the solution, as long as $NH_4^+$ is the dominant form in the Brønsted-Lowry acid–base pair with ammonia ($NH_3$).

$$NH_3 + H^+ \rightleftharpoons NH_4^+ \tag{3}$$

Alkalinization increases the amount of carbonate ($CO_3^{2-}$) in the solution:

$$H_2CO_3 \rightleftharpoons H^+ + HCO_3^- \rightleftharpoons 2\,H^+ + CO_3^{2-} \tag{4}$$

In the presence of $Ca^{2+}$, this increase in $CO_3^{2-}$ leads to a supersaturation of calcium carbonate, promoting precipitation. Thus, when modeling EICP, the key issue is predicting the distribution of ureolytic activity within the porous medium, since urease is the main agent and the prerequisite for EICP. Hence, we focus on modeling the apparent ureolytic activity as measured in the batch experiments along with its increased rate of inactivation at elevated temperatures [22,23].

## 3. Model Concept

For the sake of completeness, we explain the model's full capability. However, in this study, the terms related to two-phase effects and temperature-induced precipitation have no relevance for the given experimental conditions. Our EICP model concept accounts for two-phase, multicomponent, non-isothermal reactive transport on the scale of a representative elementary volume (REV) where flow is modeled using Darcy's Law. It includes pH-dependent dissociation reactions, temperature-dependent urease inactivation, as well as temperature-dependent enzymatically catalyzed ureolysis. Mass transfer may occur between both fluid phases by mutual dissolution of water and $CO_2$ in the gas or the aqueous phases or between the aqueous phase and the two "solid" phases adsorbed urease (au) and calcium carbonate (c) by adsorption or desorption of urease and precipitation or dissolution of calcium carbonate. We assume all calcium carbonate precipitates as calcite, as in [21]. The components considered mobile are denoted by superscripts $\kappa$: water (w), dissolved inorganic carbon ($C_{tot}$), sodium (Na), chloride (Cl), calcium (Ca), urea (u), ammonium and ammonia ($N_{tot}$), and suspended urease (su). While our model employs an REV-scale and thus a volume-averaged approach, a pore-scale representation of the relevant processes and phases is shown in Figure 1.

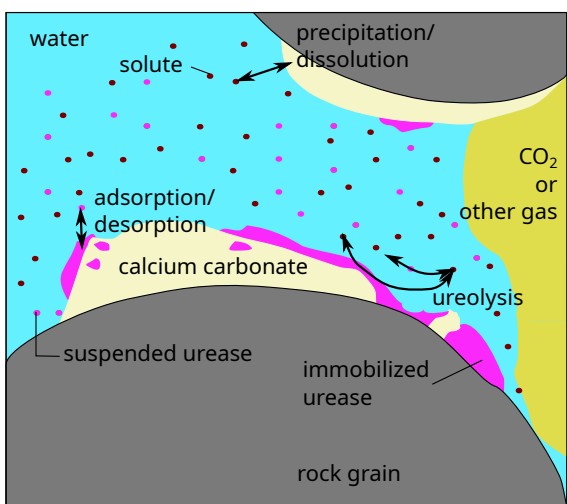

**Figure 1.** Schematic view of relevant processes and phases considered in the conceptual model, modified from [29].

The balance equations for the mobile components are solved for the primary variables $p_w$ (aqueous-phase pressure) and $x_w^\kappa$ (mole fractions of component $\kappa$ in the water phase). For the solid phases, we use their volume fractions $\phi_\lambda$. However, the gas-phase saturation $S_g = $ gas volume/pore volume is used as a primary variable instead of the mole fraction of total inorganic carbon in water $x_w^{C_{tot}}$ whenever both fluid phases are present within the same control volume. All reactive and mass-transfer processes are incorporated in the component balance Equations (5) and (6) using component-specific source and sink terms.

$$\sum_\alpha \left[ \frac{\partial}{\partial t} \left( \phi \rho_\alpha x_\alpha^\kappa S_\alpha \right) + \nabla \cdot \left( \rho_\alpha x_\alpha^\kappa \mathbf{v}_\alpha \right) - \nabla \cdot \left( \rho_\alpha D_{pm,ff}^\kappa \nabla x_\alpha^\kappa \right) \right] = q^\kappa. \tag{5}$$

Here, $t$ is time; $\phi$ is porosity; $\rho_\alpha$, $S_\alpha$, and $\mathbf{v}_\alpha$ are the density, saturation, and velocity of phase $\alpha$, respectively; $x_\alpha^\kappa$ is the mole fraction of component $\kappa$ in phase $\alpha$; $D_{pm,ff}^\kappa$ is the diffusion coefficient for component $\kappa$ in phase $\alpha$ in the porous medium; and $q^\kappa$ is the source term of component $\kappa$ due to biochemical reactions. The mass balances for the solid phases calcite and adsorbed urease contain only storage and source terms since they are assumed immobile.

$$\frac{\partial}{\partial t} \left( \phi_\lambda \rho_\lambda \right) = q^\lambda. \tag{6}$$

Here, $\phi_\lambda = $ volume$_\lambda$/total volume is the volume fraction of the solid phase $\lambda$, $\rho_\lambda$ is the solid-phase density, and $q^\lambda$ is the source term of phase $\lambda$ due to bio- or geochemical reactions. The sources and sinks due to reactions $q^\kappa$ and $q^\lambda$ are specific to the following components:

- water: $q^w = 0$
- sodium: $q^{Na} = 0$
- chloride: $q^{Cl} = 0$
- urea: $q^u = -r_u$
- ammonium and ammonia: $q^{N_{tot}} = 2r_u$
- dissolved inorganic carbon: $q^{C_{tot}} = r_{diss} - r_{prec} + r_u$
- calcium: $q^{Ca} = r_{diss} - r_{prec}$
- calcite: $q^c = r_{prec} - r_{diss}$
- suspended urease: $q^{su} = -r_i^{su} - r_a + r_d$
- adsorbed urease: $q^{au} = -r_i^{au} + r_a - r_d$

The reaction rates used to calculate the source terms are as follows:

- $r_u$: the rate of ureolysis
- $r_{diss}$ and $r_{prec}$: the rates of $CaCO_3$ dissolution and precipitation
- $r_i^{su}$ and $r_i^{au}$: the inactivation rates of suspended and adsorbed urease
- $r_a$ and $r_d$: the adsorption and desorption rates of urease

We consider ureolysis by suspended urease $r_u^{su}$ and ureolysis by adsorbed urease $r_u^{au}$. Thermal ureolysis is due to high temperatures $r_u^T$ (thermally induced calcium carbonate precipitation (TICP)).

$$r_u = r_u^{su} + r_u^{au} + r_u^T. \tag{7}$$

The reaction rates for urease-catalyzed ureolysis are modeled to be of first order with respect to urease and urea (substrate) concentration, as suggested by [22,23]; the thermal ureolysis rate is assumed to be of first order with respect to only the urea concentration, inhibited by the presence of $Ca^{2+}$ [30]. The rates are as follows:

$$r_u^{su} = \frac{k_u^e C_w^u}{M^u} C_w^{su} \phi S_w \tag{8}$$

$$r_u^{au} = \frac{k_u^e C_w^u}{M^u} \rho_{au} \, \phi_{au} \tag{9}$$

$$r_u^T = \frac{k_u^T C_w^u}{M^u} \tag{10}$$

where $C_w^u$ is the mass concentration of urea in the aqueous phase in $kg/m^3$, $M^u$ is the molar mass of urea, $\phi_{au}$ and $\rho_{au}$ are the volume fraction and the density of the crude urease source, $C_w^{su}$ is the mass concentration of suspended crude urease in the aqueous phase, $S_w$ is the water phase saturation, and $k_u^e$ and $k_u^T$ are the temperature-dependent rate coefficients for ureolysis by crude urease and elevated temperatures, respectively. Both rate coefficients are calculated using Arrhenius-type exponential relations:

$$k_u^e = c_{u,0}^e e^{\frac{c_{u,T}^e}{T}}, \tag{11}$$

$$k_u^T = c_{u,0}^T e^{\left(\frac{c_{u,T}^T}{T} - c_{u,Ca^{2+}}^T m^{Ca^{2+}}\right)}. \tag{12}$$

where $c_{u,0}^e$ and $c_{u,0}^T$ are the pre-exponential factors for enzymatic and thermal ureolysis, $T$ is the absolute temperature, $c_{u,T}^e$ and $c_{u,T}^T$ are the lumped exponents describing the temperature dependence of the rate coefficients for enzymatic and thermal ureolysis or the activation energy of ureolysis divided by the universal gas constant, $c_{u,Ca^{2+}}^T$ is an exponent describing the influence of calcium on thermal ureolysis, and $m^{Ca^{2+}}$ is the molality of calcium. We use the apparent enzymatic ureolysis rate coefficients of [23] and the thermal ureolysis rate coefficients from [30]. Note that we assume identical rate coefficients for suspended and adsorbed crude urease.

The precipitation and dissolution rates are calculated depending on the interfacial area available for the reaction as well as the saturation index $\Omega$ and, in the case of the dissolution, additionally on the molality of $H^+$. The precipitation rate of $CaCO_3$ is calculated as follows:

$$r_{prec} = k_{prec} A_{sw} (\Omega - 1)^{n_{prec}} \, ; \text{ for } \Omega \geq 1, \tag{13}$$

$$A_{sw} = A_{sw,0} \left(1 - \frac{\phi_c}{\phi_0}\right)^{\frac{2}{3}}, \tag{14}$$

$$\Omega = \frac{m^{Ca^{2+}} \gamma^{Ca^{2+}} m^{CO_3^{2-}} \gamma^{CO_3^{2-}}}{K_{sp}}, \tag{15}$$

where $k_{prec}$ and $n_{prec}$ are empirical precipitation parameters from [31], $A_{sw}$ and $A_{sw,0}$ are the current and initial interfacial areas respectively between the water phase and the solid phases, $K_{sp}$ is the calcite solubility product, and $m^{Ca^{2+}}$ and $m^{CO_3^{2-}}$ are the molalities of calcium and carbonate respectively. The activity coefficients of calcium and carbonate, $\gamma^{Ca^{2+}}$ and $\gamma^{CO_3^{2-}}$, are calculated using Pitzer equations [32–34]. The dissolution rate of $CaCO_3$ is calculated as follows:

$$r_{diss} = \left(k_{diss,1} m^{H^+} + k_{diss,2}\right) A_{cw} (\Omega - 1)^{n_{diss}} \, ; \text{ for } \Omega < 1, \tag{16}$$

$$A_{cw} = \min\left(A_{sw}, a_c \phi_c\right), \tag{17}$$

where $k_{\text{diss,1}}$, $k_{\text{diss,2}}$, and $n_{\text{diss}}$ are dissolution parameters [35,36]; $A_{\text{cw}}$ is the interfacial area of $CaCO_3$ and water; $a_{\text{c}}$ is the specific surface area of $CaCO_3$; and $\phi_{\text{c}}$ is the volume fraction of calcite. To account for ad- and desorption of the crude urease, we use first order ad- and desorption rates:

$$r_{\text{a}} = \frac{k_{\text{a}} C_{\text{w}}^{\text{su}} \phi S_{\text{w}}}{M^{\text{su}}}, \tag{18}$$

$$r_{\text{d}} = \frac{k_{\text{d}} \phi_{\text{au}} \rho_{\text{au}}}{M_{\text{au}}}. \tag{19}$$

Here, $r_{\text{a}}$ and $r_{\text{d}}$ are the ad- and desorption rates, $k_{\text{a}}$ and $k_{\text{d}}$ are the ad- and desorption rate coefficients, and $M^{\text{su}} = M_{\text{au}}$ is the molar mass of urease. The exact molar mass of our crude urease is unknown. Therefore, we balance urease in our code in mass units. To be consistent with the balance equations (Equations (5) and (6)) formulated in molar units, we introduce a dummy value $M^{\text{su}} = M_{\text{au}} = 1$ kg/mol. We use simple first order ad- and desorption rates, as it is unclear whether in our setup the urease stays inside the inactivated cells, adsorbs to cell remains, or gets completely released. Thus, we cannot justify the use of more complex adsorption or attachment kinetics published for bacterial cells, e.g., as discussed in [37]. The attachment process might be influenced by other parameters, e.g., the salinity as in [38,39].

Finally, we account for urease inactivation, assuming a first-order inactivation with an Arrhenius-type temperature dependence, again assuming as for urease activity identical rate coefficients for suspended and attached urease:

$$r_{\text{ia}}^{\text{su}} = \frac{k_{\text{ia}}^{\text{su}} C_{\text{w}}^{\text{su}} \phi S_{\text{w}}}{M^{\text{su}}}, \tag{20}$$

$$r_{\text{ia}}^{\text{au}} = \frac{k_{\text{ia}}^{\text{au}} \phi_{\text{au}} \rho_{\text{au}}}{M_{\text{au}}}, \tag{21}$$

$$k_{\text{ia}}^{\text{su}} = c_{\text{ia,0}} e^{\frac{c_{\text{ia,T}}}{T}}. \tag{22}$$

$$k_{\text{ia}}^{\text{au}} = k_{\text{ia}}^{\text{su}} + \left( \frac{r_{\text{prec}} M_{\text{c}}}{\rho_{\text{c}} (\phi_0 - \phi_{\text{c}})} \right)^{c_{\text{ia,prec}}}. \tag{23}$$

Here, $k_{\text{ia}}^{\text{su}}$ and $k_{\text{ia}}^{\text{au}}$ are the urease inactivation rate coefficients for suspended and adsorbed urease, respectively; $c_{\text{ia,0}}$ is the pre-exponential factor of an Arrhenius-type relation; $c_{\text{ia,T}}$ its lumped temperature-dependent coefficient; and $c_{\text{ia,prec}}$ is an exponent to account for preferential precipitation in the vicinity of the adsorbed urease as an additional cause of inactivation. The values for the inactivation rate coefficients $c_{\text{ia,0}}$ and $c_{\text{ia,T}}$ are taken from [23] and specific for heat-inactivated *S. pasteurii* cells.

### 3.1. Supplementary Equations

The permeability decreases due to calcite precipitation and urease adsorption calculated based on the reduction of porosity using a power law (e.g., [40]) with an exponent of three as used by both [29] and [21] for modeling MICP in the same type of porous medium as in this study:

$$\frac{K}{K_0} = \left( \frac{\phi}{\phi_0} \right)^3. \tag{24}$$

Here, $K_0$ is the initial permeability and $\phi$ and $\phi_0$ are the current and the initial porosity, respectively. The porosity $\phi$ decreases as the volume fractions of adsorbed urease and calcite increase:

$$\phi = \phi_0 - \phi_{\text{c}} - \phi_{\text{au}}. \tag{25}$$

The capillary–pressure–saturation and relative–permeability–saturation relations of Brooks and Corey [41,42] are used to calculate the capillary pressure, using an entry pressure $p_d = 10^4$ Pa and a pore-size distribution parameter $\lambda = 2$ as previously used for similar porous media [43]. The relative permeabilities of the wetting and the non-wetting phase are also calculated using the relations given by Brooks and Corey [41,42]. The impact of the calcite precipitation on capillary pressure $p_c$ is accounted for by using Leverett scaling [44] to adapt the capillary pressure of the initial porous medium $p_{c,0}$, assuming that both contact angle and surface tension do not change significantly:

$$p_c = p_{c,0} \sqrt{\frac{K_0 \phi}{K \phi_0}}. \tag{26}$$

The density and the viscosity of the $CO_2$ phase are calculated using the relation given by [45] and [46], respectively. In these calculations, the effects of the small amounts of water in the $CO_2$-phase are neglected. The density and the viscosity of the aqueous phase are calculated according to [47] as a function of salinity. Sodium, chloride, and calcium are considered to contribute to the salinity and to thus affect the aqueous phase properties.

The dissolution of $CO_2$ in the aqueous phase and the dissolution of water in $CO_2$ is calculated according to [48], with the equilibrium conditions in both phases being dependent on the pressure, temperature, and salinity of the aqueous phase. For further details on the phase composition calculations, see [49].

The temperature- and solution-composition-dependent speciation of $NH_3$ and $H_2CO_3$ produced by ureolysis (see Equations (2)–(4)) is important for modeling EICP as the precipitation rate is dependent on the activity of $CO_3^{2-}$, which is influenced by the solution's chemistry. The speciation of inorganic carbon is calculated using the apparent dissociation constants depending on ionic strength and temperature given by [50]. The temperature dependence of the speciation of $NH_4^+$ is calculated according to [51], and the effects of salinity are included using the ionic strength-dependency provided by [52], as no relation was found accounting for both salinity and high temperature. The activity of $H^+$ is calculated using the charge balance of the resulting geochemical system as well as the law of mass action for the dissociation of water. The charge balance in general requires the following:

$$\sum_{\kappa=1}^{\text{charged components}} z^\kappa m^\kappa = 0, \tag{27}$$

where $z^\kappa$ is the charge of component $\kappa$ and $m^\kappa$ is in charge of its molality. The resulting charge balance for the specific geochemical system can be written as follows:

$$0 = 2m^{Ca^{2+}} + m^{Na^+} + m^{NH_4^+} + m^{H^+} - 2m^{CO_3^{2-}} - m^{HCO_3^-} - m^{Cl^-} - m^{OH^-}. \tag{28}$$

In this equation, the molalities $m^\kappa$ of $NH_4^+$, $CO_3^{2-}$, $HCO_3^-$, and $OH^-$ are expressed as a function of $m^{H^+}$ using the laws of mass action for the dissociation reactions. The equation is then solved using an internal Newton algorithm, and the resulting activity of $H^+$ is used to calculate the molalities of the other chemical species involved in the abovementioned dissociation reactions.

## 3.2. Numerical Implementation

The model is implemented in the open-source simulator DuMu$^X$ (DUNE for Multi-Phase, Component, Scale, Physics, ...) [53] which in turn is based on DUNE (Distributed and Unified Numerics Environment), providing a framework for solving partial differential equations [54,55]. The discretization scheme used in this study is the Box method [56], a finite-volume-type approach. The implicit Euler method is applied

for time. The resulting system of equations is linearized using the Newton–Raphson method and solved using the BiCGStab solver [57]. The code is publicly available at https://git.iws.uni-stuttgart.de/dumux-pub/hommel2019a.git.

## 4. Model Calibration

Inverse modeling is used to calibrate the numerical model using the experimental results of column experiment #1 (0.33 M). The grid used for the DuMu$^X$ model is chosen such that the grid nodes match the experimental sampling locations; see Tables A1 and A2.

To this end, the developed forward model in DuMu$^X$ (see Section 3) is coupled with the Model-Independent Parameter Estimation (PEST) protocol using parameter input files. For details of the inverse modeling procedure in general, or PEST specifically, we refer interested readers to [58]. As information on the ad- and desorption rate coefficients $k_a$ and $k_d$ are virtually nonexistent, those parameters are chosen as fitting parameters. By determining the amount of crude urease and, therefore, ureolytic activity within the simulation domain with alternating injections of crude urease and mineralization medium (see Section 2) they are also expected to have a significant impact on the model predictions. All model parameters not used for calibration are given in the appendix, Table A4.

The measurements used for the calibration of the model are the urea and calcium concentrations over time at both measurement ports of the first column experiment with 0.33 M injection concentrations of the mineralization medium; see Table A1 in the appendix. All other experimental data were not used for calibration but reserved as data for validation of the calibrated model:

- the final CaCO$_3$ content along the length of both column experiments, given in Table A2 and
- the urea and calcium concentrations measured in the second (0.66 M injection concentration) column experiment; see Table A1.

## 5. Results

The output files of the model calibration (column #1) and those for the model validation (column #2) as well as the experiment data are available at https://darus.uni-stuttgart.de/dataverse/sfb1313_eicp_model_calibration.

When using only the ad- and desorption rate coefficients $k_a$ and $k_d$ as fitting parameters, the fluctuation of the concentrations can be matched qualitatively (Figures 2 and 3). The estimates for the ad- and desorption coefficients are $k_a = 3.993 \times 10^{-2}$ $^1$/s and $k_d = 8.230 \times 10^{-13}$ $^1$/s. The urea concentrations are predicted qualitatively by the calibrated model. However, some features of the experimental observations are not well reproduced by the calibrated model, such as the depletion of Ca$^{2+}$ from the mineralization solution for times of 30 min and later after the end of injection of mineralization solution. Also, the final volume fraction of precipitated calcite along the column, not used for calibration, is overestimated in the first half of the column; see Figure 4. This is somewhat similar to [20], where the model for MICP cannot predict the heterogeneity of the experimental measurements in the inlet region.

The overestimation of calcite precipitation close to the inlet suggests that, in the presented model concept (see Section 3), some of the processes might not be correctly parameterized or that some of the boundary, initial, or other experimental conditions were not reproduced in the model. After carefully reexamining the implementation of the experimental setup in the model, we are confident that we represent the experimental conditions in our simulation setup. To find out which processes might not be represented or not be correctly parameterized, we added additional parameters into the set of fitting parameters. While many more processes might be responsible for the mismatch, we chose to limit this study to parameters or processes that have been proven to be useful calibration parameters before to those processes for which

we could not find a good parameterization for our conditions in the available literature or based on experimental observations:

- The apparent ureolytic activity of the crude urease injected ($c_{u,0}^e$); see Equation (11). Also for previous MICP model calibrations, it was determined that the ureolytic activity was lower in column experiments than in fully mixed batch experiments (e.g., [21]).
- The dissociation constant for ammonia–ammonium (Ka). The current model uses a combination of the relations accounting for high temperature [51] and salinity [52]. We test whether this introduces some error by fitting a multiplier $f_{pKa}$ to the calculation of the pKa according to combined relations of [51,52] $pKa_{comb}$: $pKa = f_{pKa} \times pKa_{comb}$.
- The injected concentrations of calcium and urea $C_{Ca^{2+},inj}$ and $C_{urea,inj}$. The measurements suggest that the ratio might not have been equimolar, and less calcium is measured than expected.

The initial parameter value guesses and the set upper and lower bounds are given in Table A3 in the Appendix A. When using the injection concentrations as additional fitting parameters, the adsorption rate coefficient is estimated as $k_a = 4.459 \times 10^{-2}$ $^1/s$, an increase of 11.7% compared to the sorption-coefficients-only case. The desorption rate coefficient is estimated to be half an order of magnitude larger than for the base case, $k_d = 3.018 \times 10^{-12}$ $^1/s$. The injection concentrations are estimated as $C_{Ca^{2+},inj} = 10.64$ g/L and $C_{urea,inj} = 20.31$ g/L, which is, for calcium, only 80% of the prepared 0.33 M mineralization medium and, for urea, 101.5% of the nominal concentration of the 0.33 M injection solution ($C_{Ca^{2+},prep} = 13.3$ g/L, $C_{urea,prep} = 20.0$ g/L). The reduction in the estimated calcium concentration is approximately the residual calcium concentration predicted in the model in the case where only the ad- and desorption rate coefficients were fitted; see Figures 2 and 3.

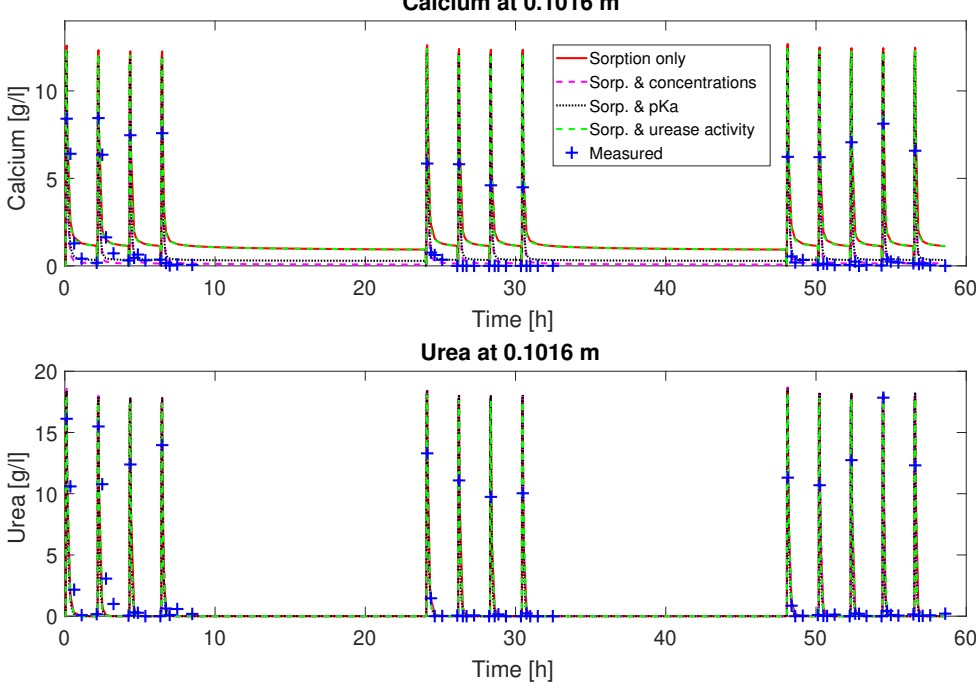

**Figure 2.** Comparison of the results of the model calibration attempts to the concentration measurements at 10.16 cm from the inlet for column experiment #1 (0.33 M mineralization medium concentration).

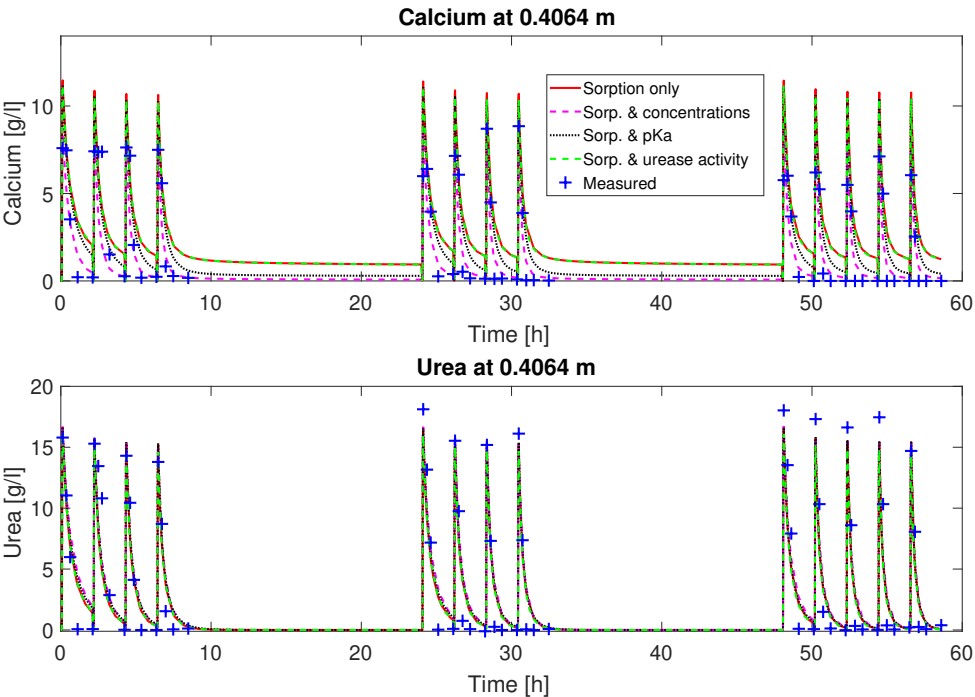

**Figure 3.** Comparison of the results of the model calibration attempts to the concentration measurements at 40.64 cm from the inlet for column experiment #1 (0.33 M mineralization medium concentration).

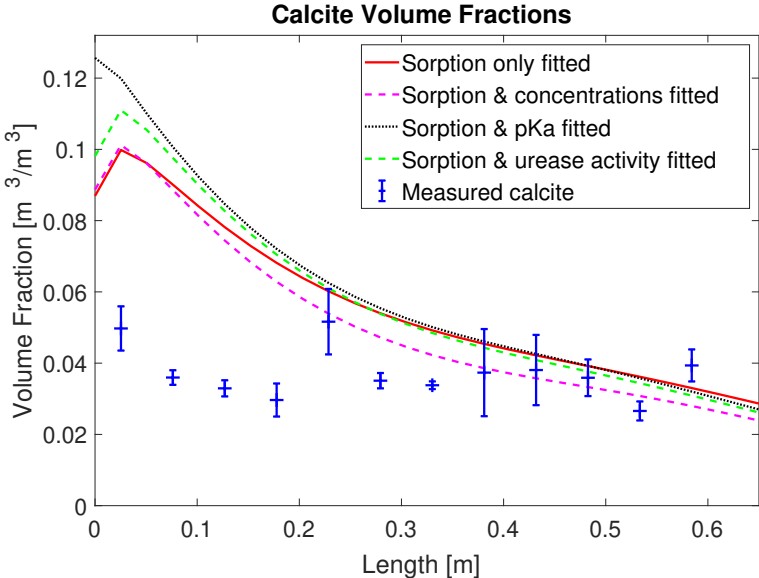

**Figure 4.** Predicted volume fractions of final calcite for the model calibration attempts compared to the experimental data for column experiment #1 (0.33 M mineralization medium concentration) for the four parameter sets fitted: The error bars represent the standard deviation calculated from triplicate measurements. Note that the calcite volume fraction was not used for calibration.

For the case with the ureolytic activity $c_{u,0}^e$ added to the set of fitting parameters, the adsorption rate coefficient increased by 8.2% to $k_a = 4.471 \times 10^{-2}$ 1/s and the desorption rate coefficient increased by orders of magnitude to $k_d = 4.534 \times 10^{-10}$ 1/s. The estimated ureolytic activity of the crude urease injected was with $c_{u,0}^e = 532.71$ m³/kg s, only 13% larger than the 462.74 m³/kg s determined in the kinetics batch experiments [23].

Including the multiplier $f_{pKa}$ to the pKa value to the set of calibration parameters resulted in the adsorption rate coefficient increasing by 8.6% to $k_a = 4.360 \times 10^{-2}$ 1/s. The desorption rate coefficient increased by more than one order of magnitude to $k_d = 1.874 \times 10^{-11}$ 1/s. The multiplier was determined to be $f_{pKa} = 1.2$, the upper limit set in the calibration, implying that the combination of the relations by [51,52] for the apparent pKa underestimates the pKa. $f_{pKa} = 1.2$ resulted in an increase of the dissociation constant Ka by 58%. The upper limit was chosen to allow for a significant change in Ka values. When comparing an extrapolation of the relation of [52] for zero salinity at 60 °C (pKa = 8.15) with the relation of [51] at 60 °C (pKa = 8.28), the difference in the predicted pKa was 0.13, resulting in a factor between both relations of 1.016. The limit set for the $f_{pKa} = 1.2$ thus allows for a more than an order of magnitude larger change of the pKa than the difference between the two relations as an estimate of the potential error. This model calibration suggests that the equilibrium between ammonium and ammonia is shifted by 0.2 pH units. As far as its impact on the conceptual model is concerned, this results in predicting that most ammonia produced by ureolysis will be converted to ammonium, consuming $H^+$ and creating calcium–carbonate precipitating conditions with less urea hydrolyzed, even at 60 °C. In a model calibration without setting an upper limit to $f_{pKa}$ (not shown here) it was estimated to a completely unrealistic value of 1.86, increasing the Ka by six orders of magnitude. During none of the other model calibrations, a parameter reached the upper or lower bounds set; see Table A3 for the values set as bounds and the initial guesses of the parameter values.

The fitted parameter values of all calibration attempts and the associated 95% confidence intervals estimated by the PEST protocol [58] are summarized in Table 1. For the adsorption rate coefficient, the 95% confidence intervals are narrow, around the estimated value ±10%, for all sets of calibration parameters. Also, except for the base case with the sorption coefficients only fitted, the estimated $k_a$ values are within the confidence intervals of the other calibration-parameter sets. Even in the sorption-coefficients-only case, $k_a = 3.993 \times 10^{-2}$ 1/s is only slightly lower than the lower limit of the confidence intervals of the calibrations set, adding either the injected concentrations or the $f_{pKa}$ to the calibration parameters.

**Table 1.** Estimated parameter values and confidence intervals as given by the Model-Independent Parameter Estimation (PEST) protocol [58] for the four investigated sets of calibration parameters calibrating to the concentration measurements of column #1 (0.33 M).

| Fit | Parameter | Estimated Value | 95% Confidence Limits | |
|---|---|---|---|---|
| | | | Lower Limit | Upper Limit |
| Sorption | $k_a$ | $3.993 \times 10^{-2}$ | $3.576 \times 10^{-2}$ | $4.410 \times 10^{-2}$ |
| coefficients only | $k_d$ | $8.230 \times 10^{-13}$ | $-6.304 \times 10^{-11}$ | $6.469 \times 10^{-11}$ |
| Sorption | $k_a$ | $4.459 \times 10^{-2}$ | $4.104 \times 10^{-2}$ | $4.813 \times 10^{-2}$ |
| coefficients and | $k_d$ | $3.018 \times 10^{-12}$ | $-1.656 \times 10^{-10}$ | $1.717 \times 10^{-10}$ |
| injection | $C_{Ca^{2+},inj}$ | 10.64 | 10.08 | 11.19 |
| concentrations | $C_{urea,inj}$ | 20.31 | 19.47 | 21.14 |
| Sorption | $k_a$ | $4.471 \times 10^{-2}$ | $3.864 \times 10^{-2}$ | $5.079 \times 10^{-2}$ |
| coefficients and | $k_d$ | $4.534 \times 10^{-10}$ | $-3.247 \times 10^{-9}$ | $4.154 \times 10^{-9}$ |
| urease activity | $c_{u,0}^e$ | 532.71 | 467.77 | 597.64 |
| Sorption | $k_a$ | $4.360 \times 10^{-2}$ | $4.017 \times 10^{-2}$ | $4.702 \times 10^{-2}$ |
| coefficients and | $k_d$ | $1.874 \times 10^{-11}$ | $-6.849 \times 10^{-10}$ | $7.224 \times 10^{-10}$ |
| pKa multiplier | $f_{pKa}$ | 1.200* | 0.8515 | 1.549 |

* Upper parameter value bound set in the calibration; see Table A3 .

In contrast, $k_d$ is not a reliably estimated parameter. It changes orders of magnitudes from one set of calibration parameters to another, being estimated to being as low as $k_d = 8.230 \times 10^{-13}$ 1/s for the sorption-coefficients-only case or as high as $4.534 \times 10^{-10}$ 1/s for the case with the sorption coefficients and the urease activity fitted. Additionally, in all sets of calibration parameters, its confidence intervals are one or more orders of magnitude larger than the estimated value itself, resulting also in unrealistic, negative lower limits of the confidence intervals.

The confidence intervals of the concentration estimates are also quite narrow with the estimated value $\pm 4.1\%$ for the urea and $\pm 5.2\%$ for the calcium injection concentration, making them relatively confidently estimated parameters. The estimated ureolytic activity coefficient $c_{u,0}^e = 532.71$ m³/kg s appears less reliable than the concentration estimates with the 95% confidence limits of 467.77 m³/kg s and 597.64 m³/kg s being at $\pm 12\%$. The ureolytic activity coefficient estimated from batch experiments is with 462.74 only slightly lower than the lower 95% confidence limit. The multiplier to the apparent pKa, $f_{pKa}$, is not very reliably estimated, as the the 95% confidence limits span from 0.851494 to 1.54851. Considering the logarithmic scale, the estimated multipliers for those limits to the actual apparent dissociation constant of ammonium and ammonia span from 0.71 to 3.54, resulting in a decrease of Ka by 29% or an increase by 254%, respectively.

Table 2 provides for all parameter calibration attempts the correlations of the estimated parameters as given by the PEST protocol [58]. The ad- and desorption rate coefficients $k_a$ and $k_d$ are not strongly correlated for any of the sets of fitted parameters. Some correlation between the ad- and desorption rate coefficients might be expected as they both determine the amount of adsorbed urease which, by its ureolytic activity, is the main driver of the overall induced $CaCO_3$ precipitation reaction. Thus, a high adsorption rate coefficient might be offset by a correspondingly high desorption rate coefficient.

When adding calibration parameters other than the ad- and desorption rate coefficients, $k_d$ correlates less with the other parameters than $k_a$, but even $k_a$ reaches a maximum at a correlation of 0.743 to the ureolytic activity, when the ureolysis activity coefficient $c_{u,0}^e$ is added to the set of fitting parameters. All other parameter correlations are less than 0.5, with the second strongest parameter correlation being 0.424 for the correlation between $C_{Ca^{2+},inj}$ and $C_{urea,inj}$ in the parameter set with the injection concentrations

being added into the set of calibration parameters. This indicates that our model calibration is not overparameterized and that the chosen calibration parameters are independent of each other.

**Table 2.** Parameter correlations as given by the PEST protocol [58] for the four investigated sets of calibration parameters calibrating to the concentration measurements of column #1 (0.33 M).

| | | $k_a$ | $k_d$ | $C_{Ca^{2+},inj}$ | $C_{urea,inj}$ | $c^e_{u,0}$ | $f_{pKa}$ |
|---|---|---|---|---|---|---|---|
| Sorption | $k_a$ | 1.0 | 0.156 | - | - | - | - |
| coefficients only | $k_d$ | 0.156 | 1.0 | - | - | - | - |
| Sorption | $k_a$ | 1.0 | 0.172 | −0.352 | −0.218 | - | - |
| coefficients and | $k_d$ | 0.172 | 1.0 | 0.105 | $8.01 \times 10^{-2}$ | - | - |
| injection | $C_{Ca^{2+},inj}$ | −0.352 | 0.105 | 1.0 | 0.424 | - | - |
| concentrations | $C_{urea,inj}$ | −0.218 | $8.01 \times 10^{-2}$ | 0.424 | 1.0 | - | - |
| Sorption | $k_a$ | 1.0 | $9.76 \times 10^{-3}$ | - | - | 0.743 | - |
| coefficients and | $k_d$ | $9.76 \times 10^{-3}$ | 1.0 | - | - | 0.135 | - |
| urease activity | $c^e_{u,0}$ | 0.743 | 0.135 | - | - | 1.0 | - |
| Sorption | $k_a$ | 1.0 | −0.129 | - | - | - | $4.64 \times 10^{-2}$ |
| coefficients and | $k_d$ | −0.129 | 1.0 | - | - | - | $6.07 \times 10^{-2}$ |
| pKa multiplier | $f_{pKa}$ | $4.64 \times 10^{-2}$ | $6.07 \times 10^{-2}$ | - | - | - | 1.0 |

Judging from the plots of predicted concentrations (Figures 2 and 3), all four sets of calibrated parameters are able to predict the urea and calcium concentrations over time qualitatively for column #1 (0.33 M). However, only the sets with a multiplier to the pKa or the injection concentrations added to the fitting parameters show the experimentally observed complete consumption of calcium.

The residuals between model predictions and experiment measurements are calculated as the sum of the squares of the difference between the model prediction $x_i^{mod}$ and the experimental observation $x_i^{exp}$:

$$r = \sum_i \left( x_i^{mod} - x_i^{exp} \right)^2. \tag{29}$$

Table 3 summarizes the residuals for all sets of data (calcium and urea concentrations at 0.1016 m and 0.4064 m, and final calcite distribution along the column) as well as the sum of the residuals for the concentrations. The sum of the residuals can be decreased by including the injection concentrations ($C_{Ca^{2+},inj}$ and $C_{urea,inj}$), the urease activity ($c^e_{u,0}$), or the multiplier to the pKa ($f_{pKa}$) to the set of calibration parameters in addition to the ad- and desorption rate coefficients $k_a$ and $k_d$. This effect is strongest for the parameters sets adding the injection concentrations or the multiplier to the pKa; see Table 3. The residual reduction for the case including the urease activity is probably related to the relatively high correlation of $k_a$ and $c^e_{u,0}$ in this case. In those two cases, the effect is mainly due to the reduction in the residuals for the calcium concentration measurements. The residuals for the urea concentration measurements do not change significantly for any of the investigated sets of fitting parameters. Adding the urease activity does not significantly reduce the sum of the residuals. The lowest total residual can be achieved when adding the concentrations to the fitting parameters, which also results in the lowest residual for the precipitated calcite volume fraction. Adding the urease activity or the multiplier to the pKa results in a higher residual for the predicted precipitated calcite volume fractions; see Table 3. Note that the calcite volume fractions were not used as observations during the calibration procedure.

In general, similar trends can be seen when using the four parameter sets, as calibrated to the concentration measurements of column #1 (0.33 M), to predict the concentrations of urea and calcium over time and the final distribution of precipitated calcite for the second column experiment (0.66 M); see

Figures 5 and 6. All fitted parameters were the same as in the simulations for column #1, except for the parameter set including the injected concentrations of the mineralization medium, for which the fitted values for column #1 (0.33 M) were doubled. This was done assuming that whatever process might have been responsible for the change in concentrations acted proportionally to the concentrations and that the prepared solutions were of double the concentration for column #2 (0.66 M). None of the data of column #2 (0.66 M) were previously used during the calibration. The most important observation is that all parameter sets can qualitatively predict the measurements, even for the changed concentrations and injection strategy, and that the four parameter sets do not result in drastically different predictions.

**Table 3.** Residuals for the data sets of the concentrations of urea and $Ca^{2+}$ concentrations in $g^2/L^2$ for column experiment #1 (0.33 M mineralization medium concentration) as given by the PEST protocol [58]: Additionally provided are the residuals for the concentration measurements of column experiment #2 (0.66 M) ($g^2/L^2$) and for both experiments the residuals for the final $CaCO_3$ distribution along the column length, which were not used for calibration.

| Data Set | Sorption Coefficients (SC) | SC & Injection Concentrations | SC & Urease Activity | SC & pKa |
|---|---|---|---|---|
| **Column #1 (0.33 M Mineralization Medium)** | | | | |
| Urea, at 0.1016 m | 264.69 | 271.19 | 280.51 | 276.06 |
| Urea, at 0.4064 m | 322.61 | 305.94 | 359.91 | 310.49 |
| $Ca^{2+}$, at 0.1016 m | 245.00 | 94.432 | 204.49 | 154.55 |
| $Ca^{2+}$, at 0.4064 m | 335.13 | 140.44 | 302.19 | 230.32 |
| Sum of concentration residuals | 1167.5 | 812.01 | 1147.1 | 971.43 |
| Final $CaCO_3$ | 0.0099 | 0.0086 | 0.0125 | 0.0157 |
| **Column #2 (0.66 M Mineralization Medium)** | | | | |
| Urea, at 0.1016 m | 36469 | 38357 | 38278 | 36960 |
| Urea, at 0.4064 m | 24922 | 22975 | 25330 | 23696 |
| $Ca^{2+}$, at 0.1016 m | 3810 | 6964 | 4015 | 4728 |
| $Ca^{2+}$, at 0.4064 m | 4932 | 3360 | 4673 | 5150 |
| Sum of concentration residuals | 70133 | 70656 | 72296 | 70535 |
| Final $CaCO_3$ | 0.0274 | 0.0227 | 0.0308 | 0.0318 |

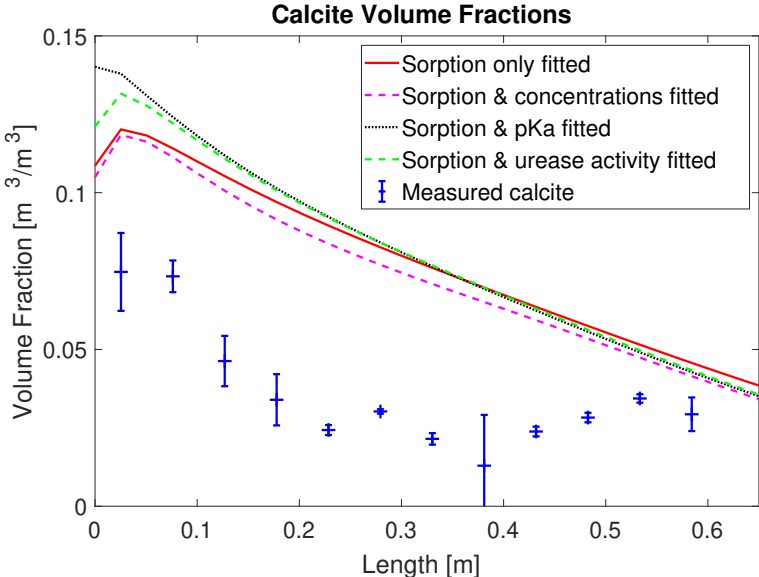

**Figure 5.** Predicted volume fractions of final calcite compared to the experimental data of column experiment #2 (0.66 M mineralization medium concentration): Note that, during this experiment, urease was only injected before every odd-numbered mineralization-medium injection. The error bars represent the standard deviation calculated from triplicate measurements. None of this experiment's data were used for calibration.

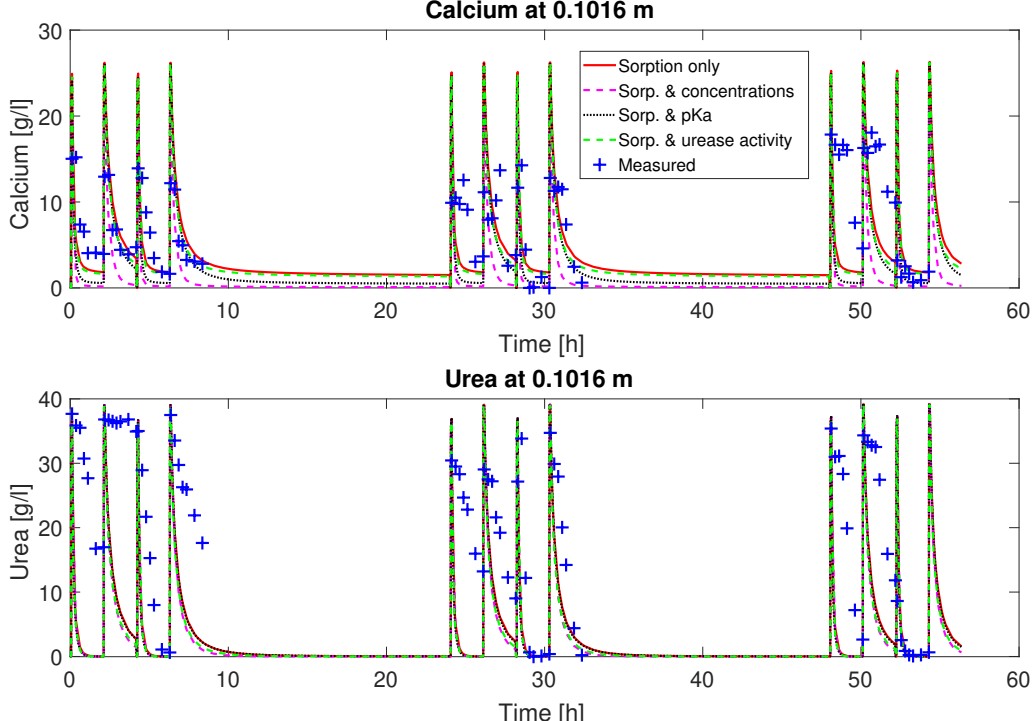

**Figure 6.** Prediction of concentration measurements at 10.16 cm from the inlet compared to the experimental data of column experiment #2 (0.66 M mineralization medium concentration): Note that, during this experiment, urease was only injected before every odd-numbered mineralization-medium injection. None of this experiment's data were used for calibration.

For all parameter sets and every sub-data set, the residuals for column #2 (0.66 M) are much larger, approximately one order of magnitude compared to those of column #1 (0.33 M); see Table 3. The increased residuals are expected, as the data were not used for calibration. The injected urea and calcium concentrations were doubled for column #2 (0.66 M), increasing the range of concentrations and thus the potential discrepancies between model predictions and measurements. This doubling of the injection concentrations might be responsible for an increase in the residuals be a factor of four when assuming a linear scaling of the difference between measured and predicted concentrations; see Equation (29). In addition, urease was added only before every odd-numbered mineralization-medium injection for column #2 (0.66 M), which may have contributed to the increased residuals. As a consequence, the model-predicted ureolysis and precipitation reactions behave differently for the odd- and the even-numbered injections, which is not for all injections the case in the experiment (Figure 6). For all calibration parameters sets, the model predicts reduced ureolytic activity and, thus, precipitation very regularly and evenly for all even-numbered mineralization-medium injections. The experimental measurements however do not show a similarly consistent, repetitive decrease of the ureolytic activity for even-numbered mineralization-medium injections. Thus, during some injection cycles, the model underpredicts ureolysis and precipitation, while in general overestimates both, leading to an overestimation of the final volume fraction of precipitated calcite (Figure 5). Interestingly, the sum of the concentration residuals for column #2 (0.66 M) is lowest for the sorption-coefficients-only case and increases for all other parameter sets in which additional parameters were calibrated. The residuals for the

final calcite of column #2 (0.66 M) behave very similarly to those of column #1 (0.33 M), as they are lowest for the parameter set including the ad- and desorption rate coefficients and the injection concentrations and the second lowest for the parameter set including only the ad- and desorption rate coefficients.

Another measure to assess the predictions of precipitated $CaCO_3$ is a comparison of the amount of the injected calcium and the amount precipitated $CaCO_3$ for each set of calibrated parameters. Therefore, we express all the injected calcium in terms of $CaCO_3$ for easier comparison with the precipitated $CaCO_3$. For column #1 (0.33 M), enough calcium to precipitate a total of 82.7 g $CaCO_3$ was injected; the case with the fitted, reduced injection concentration predicted calcium for 66.1 g of $CaCO_3$ was injected. Integrating the measured calcite along the experiment column's length results in 28.5 g for column #1 (0.33 M). In the sorption-coefficients-only case, a total of 50.3 g calcite is predicted to precipitate; in the case with the injection concentrations included, it was 45.7 g; in the case with the urease activity, it was 50.8 g; and in the case with the multiplier to pKa, it was 53.7 g. Note that, for all model calibrations, the simulation results predict that more than 60% of the injected calcium precipitates while, in the experiment, only 34% of the injected calcium precipitated.

## 6. Discussion

We used heat-inactivated cells as a urease source in this study. Since it is not yet completely clear whether the urease stays within the heat-inactivated cells or is released over time, its transport behavior cannot be exactly modeled and might be influenced by reversible and irreversible sorption. We are modeling EICP on scales larger than the molecular scale; thus, the exact state of urease within cells attached to cell residues, suspended, or absorbed is not of as much relevance for the model as the amount of urease within each control volume. Additionally, several processes could lead to changing apparent bulk ureolytic activity; e.g., sorbed urease might exhibit different activities or, when urease is released from cells, the apparent bulk ureolytic activity might rise as the limiation due to transport of urea across the cell membrane is removed. As the distribution of urease is not known for the EICP experiments, direct validation of the urease transport modeling is not possible. However, the in situ measurements of urea and $Ca^{2+}$ in the column experiments allow an indirect evaluation of the model's capabilities in predicting the urease distribution by comparing predicted and measured concentrations resulting from the ureolytic activity. Once more is known about the actual state and distribution of urease in porous media, the use of more complex and more specific sorption kinetics might be justified, e.g., as in [38,39], and might lead to better model predictions.

All model calibration attempts result in a crude urease adsorption rate coefficient of $k_a \approx 4.3 \times 10^{-2}$ 1/s and a crude urease desorption rate coefficient of $k_d < 4.534 \times 10^{-10}$ 1/s, indicating that the crude urease adsorbs mainly irreversibly to the solid surfaces. Furthermore, $k_d$ has wide confidence intervals, indicating a low model sensitivity to this parameter. Thus, for all investigated sets of calibration parameters, the desorption of crude urease is likely not significant, at least for the conditions and on the spatial scale of the column experiment used to fit the model. For conditions close to clogging, mechanical detachment of urease might remain a relevant process.

The remaining residuals for the concentration data of column #1 (see Figures 2 and 3) can to some extend be reduced by increasing the number of fitting parameters (see Table 3). This reduction is mainly due to a reduction in the residuals for the calcium concentrations. The residuals of the urea concentrations remain more or less unchanged, no matter which of the parameter sets is being fitted. Some of these residuals might be caused by the rapid ureolysis rates occurring at 60 °C. An experiment was conducted. Due to the high reaction rates, small deviations in sampling time might have led to significant differences in measured concentrations. Contrary to the findings in [21,59] for MICP and EICP, respectively, adding the urease activity to the calibrated parameters resulted in a ureolytic activity approximately equal to the

activity determined in batch experiments [22,23]. This is interesting information supporting the use of kinetic rate equations determined in batch experiments for modeling reactive transport in porous media.

The amount of precipitated $CaCO_3$ in the model seems plausible between 61% and 69% of the amount that could theoretically be precipitated based on the injected calcium. In the experiment however, only 34% of the injected calcium could be detected as $CaCO_3$. It might be that some of the $CaCO_3$ precipitated in the experiment was only loosely attached to the solid matrix or suspended long enough to be flushed out of the columns by the following injections or partly redissolved during the urease- and spacer-solution injections. In the model, it is assumed that all precipitated calcite is immobile. Potential transport of suspended precipitates is not included in the model. The model mainly overestimates the $CaCO_3$ precipitation in the inlet half of the column; see Figure 4. One reason might be that the amount of adsorbed urease and, thus, ureolytic activity in the inlet half is overestimated by the first-order adsorption kinetics used (Equation (18)), although the concentration measurements can still be reproduced with the used adsorption kinetics. This might be improved by using other, more complex adsorption kinetics as discussed in [37]. Even though less precipitates were observed in the experiment, the $Ca^{2+}$ concentrations were observed to decrease down to zero or close to zero during the no-flow periods. The calibrated model reproduces the experimentally observed complete consumption of calcium by precipitation when either the injected calcium concentration is reduced or the calculated apparent dissociation constant of ammonia–ammonium is increased. In the experiment, $Ca^{2+}$ might have formed complexes with the biomass in the system or been trapped in clogged dead-end pores as observed in [60,61], making it unavailable for both measurements and precipitation, which could explain the lower than expected concentration measurements and the low amount of precipitated $CaCO_3$ in the experiment. Some of the experimentally observed reduction in calcium might have been due to precipitation of calcium with other anions, which were not considered in the model geochemistry, such as phosphates, which were present in the cell-growth medium. The calibration would suggest that up to one sixth of the injected calcium would have precipitated as non-carbonate minerals, which seems an unlikely high amount. As discussed above, another explanation for lower calcium carbonate measurements in the columns might be the potential transport of suspended $CaCO_3$ precipitates and calcium-biomass complexes out of the column. These precipitates and calcium-biomass complexes would not have been captured as calcium in the liquid phase since samples were filtered after extraction from the column.

It is challenging to compare the experimental measurement to the model predictions for column #2, as this experiment featured a changed injection strategy, only injecting cell suspension (crude urease) before every odd-numbered mineralization-medium injection and doubling the injected concentrations of urea and calcium. Nevertheless, the model can to some extent reproduce the observed lower ureolytic activity during the mineralization periods not preceded by a urease injection. This is also reflected in the predicted volume fractions for the resulting $CaCO_3$ precipitation. Although the model overestimates the amount of precipitated $CaCO_3$ for column #2 with the 0.66 M mineralization medium concentration, the model does not predict a significantly increased volume fraction of calcite compared to column #1 with half the calcium injected. This matches qualitatively the experimental observations. The on average lower ureolytic activity and, in comparison to the injected calcium, low amount of precipitated calcite are all due to the reduced volume fraction of adsorbed urease, which is, for column #2, significantly smaller for the even-numbered injections not preceded by a urease injection; see Figure 7. The doubled injected $Ca^{2+}$ concentration of column #2 leads to more rapid inactivation of adsorbed urease due to the initially higher precipitation rate and the precipitation-rate-dependent term in the inactivation rate equation (Equation (21)). Accounting for some variability in the experimental setup, leading to variable inactivation of urease over the column length and during the course of the experiment, the model is able to reproduce the qualitative behavior of column #2. On the contrary, the active volume fraction of adsorbed urease is more or less the same or even increases during the subsequent injections over each day for column #1; see Figure 8. Thus, column #1

has relatively constant amounts of urease during each injection period and consequently similar urea and calcium concentration reductions during each of the mineralization periods; see Figures 2 and 3.

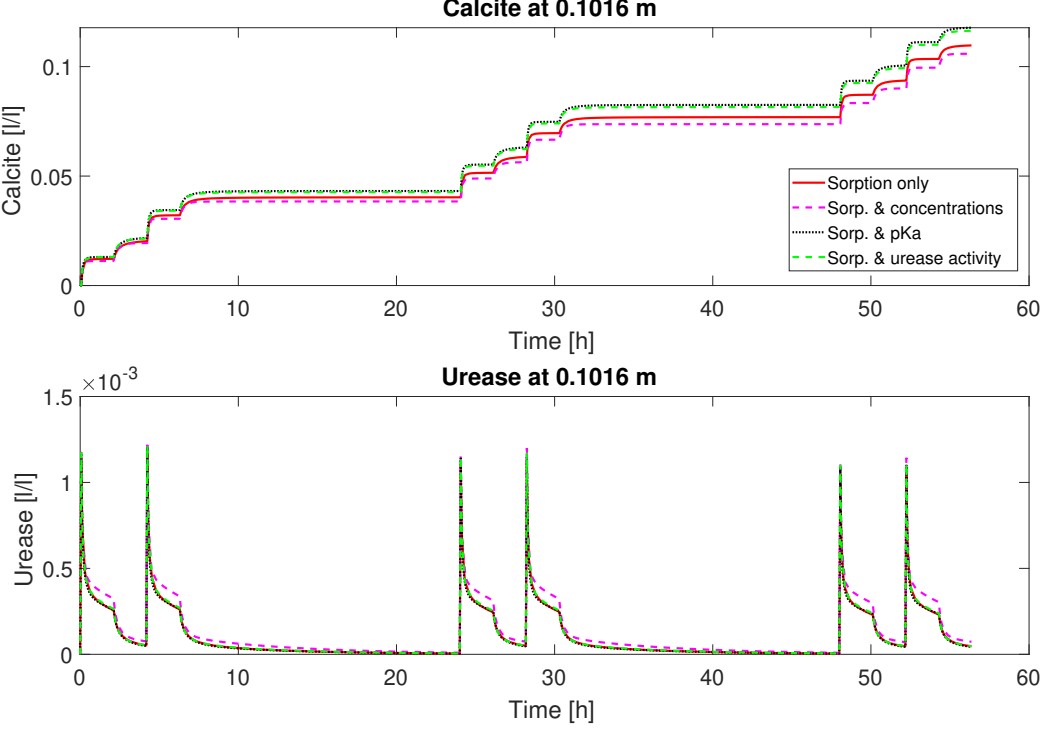

**Figure 7.** Prediction of concentration measurements at 10.16 cm from the inlet for column experiment #2 (0.66 M mineralization medium concentration): Note that, during this experiment, urease was only injected before every odd-numbered mineralization-medium injection.

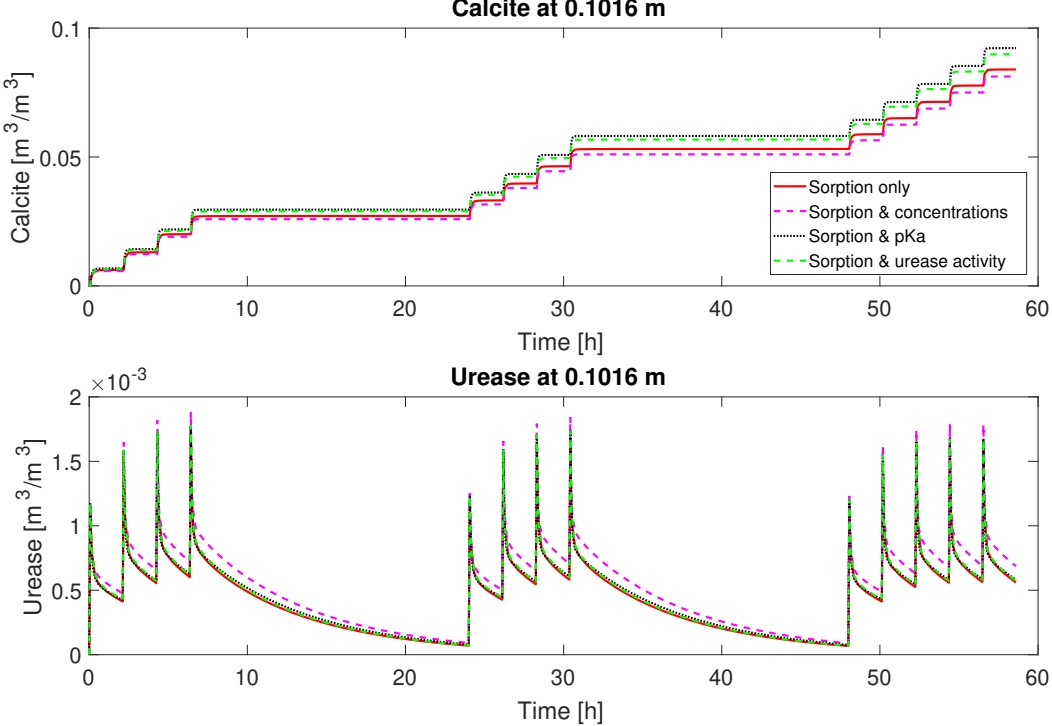

**Figure 8.** Prediction of concentration measurements at 10.16 cm from the inlet for column experiment #1 (0.33 M mineralization medium concentration).

While the residuals for the concentration measurements of column #1 (0.33 M) were somewhat reduced by adding additional fitting parameters, it is interesting to note that the residuals for the predicted volume fractions of precipitated $CaCO_3$ actually increased in most cases when more parameters than only the ad- and desorption rate coefficients were fitted. For column #2 (0.66 M), which was not used for calibration and shows larger residuals, the effect of adding more parameters to the set of calibration parameters is reversed, resulting in larger residuals for the cases with more fitting parameters. This increase is more pronounced for the case including the urease activity, which resulted in the smallest decrease of the residuals for column #1 (0.33 M). This indicates that those parameter sets where calibration parameters were added did not improve the model in terms of better representation of the physical processes, as we have hoped initially. They only give the conceptual model more flexibility by additional degrees of freedom to adapt and better match the experimental observations of column #1. One might be tempted to conclude that this falsifies the conceptual model since apparently essential processes are not captured. However, this conclusion would only be possible if all parameters, which were not in the set of estimated parameters had their "real", true values. Since this is most likely not the case, we think that this forces the calibration also for other parameters to take deviations from their "real", true values within a range that is determined by a complex matrix of correlations between parameters. For now, we propose that the calibration using only the ad- and desorption rate coefficients is a good and reasonable choice among those parameter sets that we have studied. The parameter sets with added calibration parameters do not improve the physical model as should be expected ideally when added degrees of freedom allow for better estimates of the model parameters. This was clearly not the case, and we see different potential reasons for that. It can be that the added calibration parameters were not the right ones since other parameters, which

were kept fixed, were too far from their "real" values, thus forcing the calibration parameters to take not their "real" values in order to improve the match of the model, or this discrepancy reveals problems and deficiencies in the conceptual model. In light of this, we conclude here that the calibration using only the ad- and desorption rate coefficients is the best and most reasonable choice for the existing data sets of those sets of calibration parameters investigated. As more experimental data sets on EICP become available over time, our model for EICP will improve analogously to the improvement process described in [12] for MICP modeling.

## 7. Summary and Conclusions

The developed EICP model was calibrated and validated using laboratory column experiments. Due to the increased temperatures compared to previous investigations focusing on MICP, the experiments and some of the geochemical parameterizations are associated with some level of uncertainty, but we are confident that the overall model concept and the parameterizations are robust and able to reproduce the main features of the laboratory experiments. We showed that the model is able to qualitatively predict the outcome of experiments with varied injection strategy and injected medium concentration. Using a minimal set of two calibration parameters, i.e., the ad- and desorption rate coefficients for urease, proved to be the most meaningful choice of those investigated, as adding more calibration parameters increased the residuals for the validation setup.

The presented model can now be applied to model EICP engineering problems. One application could be supporting a subsurface engineering method selection by using the presented model to compare the use of EICP to the more established MICP using any of the developed models for MICP (see Section 1) and more conventional methods of modifying subsurface properties, such as chemical grouting or cement injection, predicted by appropriate numerical models.

Further, it would be beneficial to compare the model predictions with additional experimental results of experiments conducted at various environmental or design conditions such as temperature, injection concentrations and strategies, experiment size, and complexity, similarly to the decade-long development and improvement of the (low-temperature) MICP model, which proved essential in the scale-up of the biomineralization technology [12]. Ultimately, this would help to design field applications of subsurface-modification technologies and help to choose the most promising method for a given location and its properties, comparing not only injection strategies but also different methods to induce precipitation. However, for this, the relatively complex models should be eventually simplified to be able to perform an increased number of simulations to be able to account for the imminent scenario uncertainties associated with applications in the subsurface with limited access for site investigation.

**Author Contributions:** Conceptualization, J.H., A.J.P., R.G., A.B.C., and H.C.; methodology, J.H. (modeling), A.A., and Z.F. (experiments); software, J.H.; validation, J.H., A.J.P., and R.G.; formal analysis, J.H.; investigation, J.H. (modeling), A.A., and Z.F. (experiments); resources, J.H., A.J.P., and R.G.; data curation, J.H.; writing—original draft preparation, J.H.; writing—review and editing, J.H., R.H., and H.C.; visualization, J.H.; supervision, R.H. and H.C.; project administration, J.H.; funding acquisition, J.H., A.J.P., and R.G. All authors have read and agreed to the published version of the manuscript.

**Funding:** This research was funded by the Deutsche Forschungsgemeinschaft (DFG, German Research Foundation)—project number HO 6055/1-1. The experimental work was funded by the U.S. Department of Energy, Office of Fossil Energy under the DOE award number DE-FE0026513, "Wellbore Leakage Mitigation Using Advanced Mineral Precipitation Strategies".

**Acknowledgments:** We further thank the Deutsche Forschungsgemeinschaft (DFG, German Research Foundation) for supporting this work by funding SFB 1313, project number 327154368. The authors acknowledge the Center for Biofilm Engineering and the Imaging and Chemical Analysis Laboratory at Montana State University part of the Montana Nanotechnology Facility, an NNCI facility supported by NSF grant ECCS-1542210.

**Conflicts of Interest:** The authors declare no conflict of interest.

## Abbreviations

The following abbreviations are used in this manuscript:

EICP      Enzymatically induced calcium carbonate precipitation
MICP     Microbially induced calcium carbonate precipitation
OD        Optical density
OD600nm  Optical density at a wavelength of 600 nm

## Appendix

**Table A1.** Measured concentrations of urea and calcium for the 0.33 M and the 0.66 M column experiments at 0.1016 m and 0.4064 m.

| | Column #1, 0.33 M | | | | | Column #2, 0.66 M | | | |
|---|---|---|---|---|---|---|---|---|---|
| **Time (s)** | **Calcium (g/L)** | | **Urea (g/L)** | | **Time (s)** | **Calcium (g/L)** | | **Urea (g/L)** | |
| | **0.1016 m** | **0.4064 m** | **0.1016 m** | **0.4064 m** | | **0.1016 m** | **0.4064 m** | **0.1016 m** | **0.4064 m** |
| 493 | 8.4103 | 7.5902 | 16.120 | 15.795 | 433 | 15.026 | 11.964 | 37.674 | 37.801 |
| 1333 | 6.4059 | 7.4667 | 10.598 | 11.043 | 1333 | 15.198 | 10.822 | 35.864 | 35.851 |
| 2233 | 1.2835 | 3.5200 | 2.1636 | 5.9831 | 2233 | 7.3766 | 4.6683 | 35.513 | 33.458 |
| 4033 | 0.4103 | 0.2133 | 0.0867 | 0.0655 | 3133 | 6.5543 | 6.4967 | 30.735 | 29.084 |
| 7633 | 0.1758 | 0.2021 | 0.1664 | 0.0732 | 4033 | 4.0492 | 2.2950 | 27.679 | 26.692 |
| 8126 | 8.4454 | 7.4105 | 15.492 | 15.292 | 5833 | 4.1066 | 2.2422 | 16.743 | 15.914 |
| 8966 | 6.3531 | 7.3656 | 10.781 | 13.451 | 7633 | 3.9536 | 2.0488 | 16.966 | 8.0715 |
| 9866 | 1.6293 | 7.3881 | 3.0684 | 10.807 | 7844 | 12.941 | 10.681 | 36.802 | 36.297 |
| 11,666 | 0.7150 | 1.5158 | 1.0009 | 2.8624 | 8744 | 13.152 | 9.9952 | 36.726 | 37.105 |
| 15,266 | 0.3106 | 0.2695 | 0.0492 | 0.0160 | 9644 | 6.7264 | 6.4440 | 36.522 | 35.941 |
| 15,759 | 7.4667 | 7.6295 | 12.389 | 14.301 | 10,544 | 6.8029 | 5.7759 | 36.246 | 34.492 |
| 16,599 | 0.4571 | 7.1579 | 0.3118 | 10.441 | 11,444 | 4.4317 | 2.7696 | 36.493 | 34.357 |
| 17,499 | 0.6447 | 2.0547 | 0.2789 | 4.1128 | 13,244 | 3.9154 | 2.1983 | 36.808 | 34.006 |
| 19,299 | 0.3165 | 0.1853 | 0 | 0 | 15,044 | 4.7376 | 3.0685 | 34.932 | 33.134 |
| 22,899 | 0.3575 | 0.2358 | 0 | 0 | 15,477 | 13.917 | 13.652 | 35.011 | 35.378 |
| 23,392 | 7.5839 | 7.4947 | 13.969 | 13.794 | 16,377 | 12.788 | 12.070 | 28.926 | 33.159 |
| 24,232 | 0.1641 | 5.5860 | 0.6306 | 8.7075 | 17,277 | 8.7917 | 5.8111 | 21.692 | 32.270 |
| 25,132 | 0.0234 | 0.8309 | 0.1008 | 1.5402 | 18,177 | 6.4396 | 5.7935 | 15.277 | 30.533 |
| 26,932 | 0.0821 | 0.2807 | 0.5884 | 0.0503 | 19,077 | 3.4755 | 3.8948 | 7.9859 | 24.323 |
| 30,532 | 0.0586 | 0.1740 | 0.1852 | 0.1189 | 20,877 | 1.8692 | 3.4904 | 1.0803 | 16.576 |

**Table A1.** *Cont.*

| | Column #1, 0.33 M | | | | | Column #2, 0.66 M | | | |
|---|---|---|---|---|---|---|---|---|---|
| Time (s) | Calcium (g/L) | | Urea (g/L) | | Time (s) | Calcium (g/L) | | Urea (g/L) | |
| | 0.1016 m | 0.4064 m | 0.1016 m | 0.4064 m | | 0.1016 m | 0.4064 m | 0.1016 m | 0.4064 m |
| 86,893 | 5.8474 | 5.9891 | 13.298 | 18.116 | 22,677 | 1.6397 | 2.4180 | 0.6187 | 10.411 |
| 87,733 | 0.7560 | 6.4027 | 1.4621 | 13.161 | 22,888 | 12.188 | 13.006 | 37.493 | 37.030 |
| 88,,633 | 0.6186 | 3.9592 | 0 | 7.1720 | 23,788 | 11.440 | 12.508 | 33.533 | 35.311 |
| 90433 | 0.3436 | 0.2639 | 0.0205 | 0.0243 | 24,688 | 5.4431 | 5.7990 | 29.735 | 34.519 |
| 94,033 | 0 | 0.3946 | 0.0532 | 0.0920 | 25,588 | 4.9626 | 5.6566 | 26.280 | 33.116 |
| 94,526 | 5.8089 | 7.1483 | 11.087 | 15.532 | 26,488 | 3.2541 | 3.3787 | 25.938 | 34.069 |
| 95,366 | 0 | 6.0707 | 0 | 9.7530 | 28,288 | 3.0762 | 3.3253 | 21.911 | 31.667 |
| 96,266 | 0 | 0.5252 | 0 | 0.7502 | 30,088 | 2.8092 | 3.4499 | 17.628 | 30.767 |
| 98,066 | 0 | 0.1497 | 0.0672 | 0.0694 | 86,833 | 9.9048 | 13.581 | 30.441 | 36.146 |
| 101,666 | 0 | 0.1279 | 0 | 0 | 87,733 | 10.500 | 13.363 | 29.496 | 35.616 |
| 102,159 | 4.6048 | 8.6993 | 9.7429 | 15.190 | 88,633 | 9.7642 | 14.034 | 28.307 | 33.595 |
| 102,999 | 0 | 4.4871 | 0.0065 | 7.3140 | 89,533 | 12.547 | 12.856 | 24.672 | 32.854 |
| 103,899 | 0 | 0.1224 | 0.1512 | 0.2588 | 90,433 | 9.0944 | 13.363 | 22.805 | 32.289 |
| 105,699 | 0 | 0.1388 | 0 | 0 | 92,233 | 3.0418 | 6.0016 | 15.967 | 29.928 |
| 109,299 | 0 | 0.0898 | 0 | 0 | 94,033 | 3.6702 | 6.1648 | 13.209 | 26.141 |
| 109,792 | 4.4893 | 8.8463 | 10.037 | 16.118 | 94,244 | 11.133 | 12.565 | 29.039 | 36.931 |
| 110,632 | 0 | 3.8830 | 0.2211 | 7.3681 | 95,144 | 7.9368 | 11.006 | 27.492 | 35.078 |
| 111,532 | 0 | 0.0408 | 0 | 0.0514 | 96,044 | 8.1022 | 11.949 | 27.202 | 33.737 |
| 113,332 | 0 | 0.0463 | 0 | 0 | 96,944 | 10.186 | 12.185 | 21.601 | 31.292 |
| 116,932 | 0 | 0.0245 | 0 | 0.0784 | 97,844 | 13.690 | 12.112 | 19.216 | 30.983 |
| 173,293 | 6.2306 | 5.7627 | 11.313 | 18.026 | 99,644 | 2.5787 | 6.0560 | 12.279 | 28.017 |
| 174,133 | 0.5417 | 6.0133 | 0.8575 | 13.534 | 101,444 | 3.8190 | 7.8511 | 9.0211 | 26.746 |
| 175,033 | 0.1917 | 3.6827 | 0 | 7.9176 | 101,877 | 11.658 | 13.381 | 27.157 | 25.066 |
| 176,833 | 0.3361 | 0.2267 | 0.0406 | 0.0973 | 102,777 | 14.268 | 11.060 | 33.837 | 24.621 |
| 180,433 | 0.0528 | 0 | 0.0986 | 0.0705 | 103,677 | 4.4475 | 12.928 | 12.197 | 30.728 |
| 180,926 | 6.2139 | 6.2000 | 10.694 | 17.313 | 104,577 | 0 | 12.402 | 0.6363 | 27.383 |
| 181,766 | 0.1194 | 5.2400 | 0.0309 | 10.331 | 105,477 | 0.0816 | 12.856 | 0 | 23.421 |
| 182,666 | 0.1139 | 0.4240 | 0.0164 | 1.4891 | 107,277 | 1.2657 | 5.9472 | 0.0819 | 14.517 |
| 184,466 | 0.0472 | 0 | 0.1324 | 0.1240 | 109,077 | 0 | 2.7196 | 0.3753 | 3.8747 |
| 188,066 | 0.0528 | 0 | 0 | 0 | 109,288 | 12.793 | 12.793 | 34.720 | 36.205 |
| 188,559 | 7.0639 | 5.4907 | 12.744 | 16.626 | 110,188 | 11.290 | 12.492 | 29.893 | 35.200 |
| 189,399 | 0.2417 | 3.9760 | 0.2098 | 8.6001 | 111,088 | 11.741 | 11.797 | 27.937 | 32.809 |
| 190,299 | 0 | 0 | 0.1953 | 0.3337 | 111,988 | 11.478 | 12.417 | 20.043 | 30.373 |
| 192,099 | 0.0583 | 0 | 0.0213 | 0.0437 | 112,888 | 7.3804 | 9.8049 | 14.211 | 28.924 |
| 195,699 | 0.0250 | 0 | 0 | 0.0080 | 114,688 | 2.4563 | 5.8393 | 4.3939 | 21.521 |
| 196,192 | 8.1250 | 7.1173 | 17.843 | 17.460 | 116,488 | 0.6145 | 5.5010 | 0.2067 | 18.769 |
| 197,032 | 0.3750 | 4.9893 | 0.4128 | 10.331 | 173,233 | 17.845 | 13.674 | 35.395 | 36.931 |
| 197,932 | 0.2472 | 0 | 0.1034 | 0.3783 | 174,133 | 16.664 | 12.900 | 31.000 | 34.762 |
| 199,332 | 0.1472 | 0 | 0.1469 | 0.2088 | 175,933 | 16.629 | 11.658 | 28.339 | 33.452 |
| 203,825 | 6.5806 | 6.0453 | 12.318 | 14.699 | 176,833 | 16.039 | 11.513 | 19.897 | 31.704 |
| 204,665 | 0.0972 | 2.5360 | 0.0200 | 8.0608 | 178,633 | 7.5848 | 4.1576 | 7.1984 | 22.219 |
| 205,565 | 0.1528 | 0 | 0.0314 | 0.2477 | 180,433 | 4.5989 | 4.6415 | 2.6259 | 19.619 |
| 207,365 | 0.0361 | 0 | 0.0701 | 0.0656 | 180,644 | 16.282 | 12.513 | 34.336 | 36.327 |
| 210,965 | 0 | 0 | 0.2067 | 0.3888 | 181,544 | 15.709 | 12.738 | 33.324 | 36.922 |
| | | | | | 182,444 | 18.070 | 11.303 | 32.799 | 36.254 |
| | | | | | 183,344 | 16.473 | 10.335 | 32.528 | 35.833 |
| | | | | | 184,244 | 16.682 | 9.8191 | 27.440 | 33.379 |
| | | | | | 186,044 | 11.196 | 4.8673 | 15.926 | 25.263 |
| | | | | | 187,844 | 9.9284 | 4.4480 | 11.827 | 24.050 |
| | | | | | 188,277 | 3.2101 | 0 | 8.6156 | 0.0824 |
| | | | | | 189,177 | 1.2484 | 0 | 2.5439 | 0.1648 |
| | | | | | 190,077 | 2.5157 | 0 | 0.8573 | 0.9430 |
| | | | | | 190,977 | 1.6650 | 0 | 0.2389 | 0.3387 |
| | | | | | 191,877 | 0.6581 | 0 | 0.0796 | 0.4303 |
| | | | | | 193,677 | 0.9012 | 0 | 0.2460 | 0.2609 |
| | | | | | 195,477 | 1.8907 | 0 | 0.6442 | 0.7553 |

**Table A2.** Final calcite volume fraction distribution over column length of the 0.33 M and 0.66 M column experiments.

| Distance (m) | 0.33 M | 0.66 M |
|---|---|---|
| 0.0254 | 0.0497 | 0.0603 |
| 0.0762 | 0.0360 | 0.0585 |
| 0.1270 | 0.0329 | 0.0423 |
| 0.1778 | 0.0297 | 0.0301 |
| 0.2286 | 0.0516 | 0.0200 |
| 0.2794 | 0.0351 | 0.0257 |
| 0.3302 | 0.0338 | 0.0174 |
| 0.3810 | 0.0373 | 0.0182 |
| 0.4318 | 0.0381 | 0.0182 |
| 0.4826 | 0.0359 | 0.0225 |
| 0.5334 | 0.0266 | 0.0286 |
| 0.5842 | 0.0394 | 0.0242 |

**Table A3.** Initial parameter values and upper and lower parameter bounds set for calibration with the PEST protocol [58] for the four investigated sets of calibration parameters.

| Fit | Parameter | Initial Guess | Lower Bound | Upper Bound |
|---|---|---|---|---|
| Sorption coefficients only | $k_a$ | $4.0 \times 10^{-2}$ | $1.0 \times 10^{-5}$ | $1.0 \times 10^{0}$ |
| | $k_d$ | $2.0 \times 10^{-10}$ | $1.0 \times 10^{-15}$ | $1.0 \times 10^{-3}$ |
| Sorption coefficients and injection concentrations | $k_a$ | $4.0 \times 10^{-2}$ | $1.0 \times 10^{-5}$ | $1.0 \times 10^{0}$ |
| | $k_d$ | $2.0 \times 10^{-10}$ | $1.0 \times 10^{-15}$ | $1.0 \times 10^{-3}$ |
| | $C_{Ca^{2+},inj}$ | 10.0 | 1 | 13.3 |
| | $C_{urea,inj}$ | 20.0 | 1 | 25 |
| Sorption coefficients and urease activity | $k_a$ | $4.0 \times 10^{-2}$ | $1.0 \times 10^{-5}$ | $1.0 \times 10^{0}$ |
| | $k_d$ | $2.0 \times 10^{-10}$ | $1.0 \times 10^{-15}$ | $1.0 \times 10^{-3}$ |
| | $c_{u,0}^{e}$ | 463 | 1 | 1000 |
| Sorption coefficients and pKa multiplier | $k_a$ | $4.0 \times 10^{-2}$ | $1.0 \times 10^{-5}$ | $1.0 \times 10^{0}$ |
| | $k_d$ | $2.0 \times 10^{-10}$ | $1.0 \times 10^{-15}$ | $1.0 \times 10^{-3}$ |
| | $f_{pKa}$ | 1.0 | 0.8 | 1.2 |

**Table A4.** Model parameter values used and calibrated for the presented EICP model.

| Parameter | Unit | Value | Reference |
|---|---|---|---|
| $T$ | K | 333.15 | Measured |
| $\phi_0$ | - | 0.345 | Measured |
| $\phi_{crit}$ | - | 0 | [29] |
| $K_0$ | $m^2$ | $2 \times 10^{-10}$; | [29] |
| $\rho_c$ | kg/m³ | 2710 | - |
| $\rho_{au}$ | kg/m³ | 1100 | - |
| $D_w$ | m²/s | $1.587 \times 10^{-9}$ | [62] |
| $A_{sw,0}$ | m²/m³ | 5000 | [29] |
| $a_c$ | m²/m³ | 20000 | [29] |
| $k_{prec}$ | mol/s m² | $1.5 \times 10^{-10}$ | [31] |
| $n_{prec}$ | - | 3.27 | [31] |
| $k_{diss,1}$ | $kg_{H_2O}/m^3$ | $8.9 \times 10^{-1}$ | [35] |
| $k_{diss,2}$ | mol/s m² | $6.5 \times 10^{-7}$ | [35] |
| $n_{diss}$ | - | 1 | [63] |
| $c_{u,0}^e$ | m³/kg s | 462.74<br>532.71 $^{ua}$ | [23]<br>Calibration $^{ua}$ |
| $c_{u,T}^e$ | K | $-4263.108$ | [23] |
| $c_{ia,0}$ | 1/s | $1.3340 \times 10^{23}$ | [23] |
| $c_{ia,T}$ | K | $-21140$ | [23] |
| $c_{ia,prec}$ | - | 0.67 | Estimated |

**Table A4.** *Cont.*

| Parameter | Unit | Value | Reference |
|---|---|---|---|
| $k_a$ | 1/s | $3.99 \times 10^{-2}$ $^s$<br>$4.46 \times 10^{-2}$ $^c$<br>$4.47 \times 10^{-2}$ $^{ua}$<br>$4.36 \times 10^{-2}$ $^{pk}$ | Calibration $^s$<br>Calibration $^c$<br>Calibration $^{ua}$<br>Calibration$^{pk}$ |
| $k_d$ | 1/s | $8.23 \times 10^{-13}$ $^s$<br>$3.02 \times 10^{-12}$ $^c$<br>$4.53 \times 10^{-10}$ $^{ua}$<br>$1.87 \times 10^{-11}$ $^{pk}$ | Calibration $^s$<br>Calibration $^c$<br>Calibration $^{ua}$<br>Calibration $^{pk}$ |
| $c_{u,0}^T$ | 1/s | $1.3438 \times 10^6$ | [30] |
| $c_{u,T}^T$ | K | $-9945$ | [30] |
| $c_{u,Ca^{2+}}^T$ | $kg_{H_2O}/mol$ | 0.5 | [30] |
| $C_{Ca^{2+},inj}$ | kg/m³ | 13.3<br>10.64 $^c$ | Measured<br>Calibration $^c$ |
| $C_{urea,inj}$ | kg/m³ | 20.0<br>20.31 $^c$ | Measured<br>Calibration $^c$ |
| $f_{pKa}$ | - | 1<br>1.2 $^{pk}$ | Assumed<br>Calibration $^{pk}$ |

The different calibrated parameter values are shown in Table A4 but repeated here for the sake of completeness. The respective parameter set for which the value was obtained marked by the superscripts: $^s$ sorption coefficients only ($k_a$ and $k_d$), $^c$ sorption coefficients and injection concentrations ($k_a$, $k_d$, $C_{Ca^{2+},inj}$, and $C_{urea,inj}$), $^{ua}$ sorption coefficients and urease activity ($k_a$, $k_d$, and $c_{u,0}^e$), $^{pk}$ sorption coefficients, and pKa multiplier ($k_a$, $k_d$, and pKa multiplier).

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
