# Peer review of "A Numerical Model for Enzymatically Induced Calcium Carbonate Precipitation"

_applsci, doi:10.3390/app10134538_

Round 1

Reviewer 1 Report

With much pleasure and interest I have read this manuscript. It is of high quality, reads well and is really at the frontier of current knowledge, regarding the numerical reactive transport modeling of Enzymatic or Microbial Induced Carbonate Precipitation. Although other, perhaps similar papers on this topic (including some by the same authors) are already existing, the novel aspects in this paper are the treatment and simulation of the process at elevated temperatures. The combination of original experimental results and numerical simulation, including calibration and validation is useful as it provides insight in the strengths and weaknesses of the numerical model.

As I have indicated in the annotated manuscript file some more details could be provided on the experimental methodology (like what is the flow rate, flow direction, the activity of the injected bacteria, flushed volume, system volume (total porosity including tubing and filter stones etc.) and I do not fully agree with all the selected equations (for hydrolyis kinetics I would have at least selected Michaelis-Menten equation), some of the selected parameters for the calibration (e.g. the precipitation rate coefficient and crystal surface area are typically not fixed and may stall the precipitation) and some of the interpretation of the results.

The main concern relates to the conclusions. In particular, the statement that "Using only two calibration parameters, i.e. the ad- and desorption rate coefficients for urease, proved to be the most meaningful choice to qualitatively predict the outcome of experiments" does only seem to make sense for the calibration variables considered. However, considering the large discrepancy between the average theoretically expected (66.1 to 82.7g), simulated (45.7-53.7g) and measured (28.5g) amounts of calcium carbonate at the end of the experiment, and the limited capacity to predict the substrate concentrations in the 0.66 M column (which was not used for calibration) it does not seem the model is ready yet to predict the outcome of experiments correctly, particular when aiming for larger scale and more complex treatment conditions. Clearly some mechanisms affecting the process kinetics are not yet appropriately incorporated. However, the model presented in this paper has the flexibility and capacity to be revised or extended to include alternative process variables. The fact that all the code and results are made available could help the scientific community to tackle this challenge.

Detailed comments are provided annotated manuscript. Note that most of these comments are optional suggestions for improvements as they are still part of the scientific debate.

Reviewer 2 Report

see attached file
